# Solid Lipid Nanoparticles: Multitasking Nano-Carriers for Cancer Treatment

**DOI:** 10.3390/pharmaceutics15030831

**Published:** 2023-03-03

**Authors:** Júlia German-Cortés, Mireia Vilar-Hernández, Diana Rafael, Ibane Abasolo, Fernanda Andrade

**Affiliations:** 1Drug Delivery & Targeting Group, Vall d’Hebron Institut de Recerca, Universitat Autònoma de Barcelona (UAB), 08035 Barcelona, Spain; 2Networking Research Centre for Bioengineering, Biomaterials, and Nanomedicine (CIBER-BBN), Instituto de Salud Carlos III, 28029 Madrid, Spain; 3Functional Validation & Preclinical Research (FVPR), U20 ICTS Nanbiosis, Vall d’Hebron Institut de Recerca (VHIR), Universitat Autònoma de Barcelona (UAB), 08035 Barcelona, Spain; 4Servei de Bioquímica, Hospital Universitari Vall d’Hebron, 08035 Barcelona, Spain; 5Departament de Farmàcia i Tecnologia Farmacèutica i Fisicoquímica, Facultat de Farmàcia i Ciències de l’Alimentació, Universitat de Barcelona (UB), 08028 Barcelona, Spain

**Keywords:** solid lipid nanoparticles, nanomedicine, cancer therapy, cancer diagnostics, drug delivery, targeted therapy

## Abstract

Despite all the advances seen in recent years, the severe adverse effects and low specificity of conventional chemotherapy are still challenging problems regarding cancer treatment. Nanotechnology has helped to address these questions, making important contributions in the oncological field. The use of nanoparticles has allowed the improvement of the therapeutic index of several conventional drugs and facilitates the tumoral accumulation and intracellular delivery of complex biomolecules, such as genetic material. Among the wide range of nanotechnology-based drug delivery systems (nanoDDS), solid lipid nanoparticles (SLNs) have emerged as promising systems for delivering different types of cargo. Their solid lipid core, at room and body temperature, provides SLNs with higher stability than other formulations. Moreover, SLNs offer other important features, namely the possibility to perform active targeting, sustained and controlled release, and multifunctional therapy. Furthermore, with the possibility to use biocompatible and physiologic materials and easy scale-up and low-cost production methods, SLNs meet the principal requirements of an ideal nanoDDS. The present work aims to summarize the main aspects related to SLNs, including composition, production methods, and administration routes, as well as to show the most recent studies about the use of SLNs for cancer treatment.

## 1. Introduction

Cancer treatment, especially in advanced stages, is a high-priority unmet clinical need since, until today, the high toxicity of chemotherapeutic treatment, their lack of specificity to cancer cells, and the acquired drug resistance hamper a successful cancer treatment. The oncology research field has experienced great benefits from nanotechnology. Nanotechnology-based drug delivery systems (nanoDDS) brought solutions to the major drawbacks related to conventional chemotherapy, namely: (i) drug protection from degradation, (ii) improving the solubility of hydrophobic drugs, (iii) increase of circulation time, (iv) reducing toxicity, (v) decreasing the required dose, and (vi) facilitating the overcoming of biological barriers. In summary, by encapsulating the chemotherapeutic drugs into nanoDDS, it is possible to improve its therapeutic index. Moreover, nanoDDS allow the safe and efficient delivery of genetic material and other biomolecules that could not reach the cells on their own [1].

A wide range of nanoDDS types have been reported in the last decades with countless different compositions and preparation methods. Among them, solid lipid nanoparticles (SLNs), constituted by a solid lipidic core at body temperature, called the attention of multiple research groups due to their high versatility regarding different cargos and administration routes, high stability in physiological conditions, low cost of materials, and easy production [2]. SLNs were first developed in the early 1990s as an alternative colloidal delivery system to overcome the limitations of the most common formulations at the time: emulsions, liposomes, and polymeric nanoparticles [3]. The traditional colloidal carriers deal with problems such as poor stability, polymer toxicity and degradation, high cost, and difficulties in scaling-up the production [4]. The solid lipid core of SLNs was a great improvement in the nanoDDS stability issues. Moreover, the possibility of using physiologically biocompatible products in SLN composition and their scalable production methods are highly important parameters in the development of a new nanoDDS.

## 2. SLN Properties

SLNs comprise a lipid core and are solid at room and body temperatures, surrounded by a surfactant or emulsifier, stabilizing the core by lowering the interfacial tension with the aqueous media [5]. Sometimes co-surfactants are also included in the formulation to further reduce the interfacial tension and increase the stability. The main components of SLNs used as lipids, surfactants, and co-surfactants and some examples and commercial names are described in Table 1 [4,5,6]. When SLNs are applied to gene therapy, cationic lipids are used, including, among others, n-[1-(2,3-dioleoyloxy)propyl]-n,n,n-trimethylammonium chloride (DOTAP), cetrimide (CTAB), cetylpyridinium chloride (CPC), or benzalkonium chloride [7].

The components of the formulation must be chosen based on the desired application (Figure 1). For example, only excipients approved for parenteral administration must be used for intravenous SLNs. Also, the compatibility between the drugs and the lipids/surfactants must be considered. Lipids that promote a sustained release of the drugs for days must not be chosen for applications that require a small permanence of SLNs in the body, such as oral administration.

Coating materials, viscosity enhancers, antioxidants, absorption-enhancing agents, preservatives, adhesives, and other excipients can also be added to the final formulation based on the drugs and the desired application [7].

The size of the SLNs can be adapted accordingly to the formulation components and production method, varying from 50–1000 nm. In addition, the loading capacity of hydrophobic and hydrophilic drugs and their release profile depends on SLNs composition and manufacturing [4,8]. The different techniques that can be applied for SLNs characterization were previously reviewed elsewhere [7].

SLNs’ surfaces can be functionalized with polyethylene glycol (PEG) coating to increase the efficiency of drug and gene delivery to target cells and tissues, improve the systemic circulation time, and decrease immunogenicity. The PEG coating shields the surface from aggregation, opsonization, and phagocytosis, prolonging systemic circulation time [9]. Moreover, the therapeutic effect is potentially more efficient when the SLNs selectively deliver the drug to its specific site of action with active targeting. For this, the SLN surface is functionalized with ligands (targeting moieties) that can selectively recognize overexpressed receptors on the surface of cancer cells and, ultimately, be translocated inside the cells. Consequently, the selective delivery of the pharmacologically active compounds to the tumor can reduce the toxicity and harmful side effects on other healthy cells [2]. This is being explored as a multifunctional/multitasking platform for efficient drug or gene delivery and as a diagnostic tool.

SLNs present some advantages, such as the capacity to load both hydrophobic and hydrophilic drugs, biocompatibility, low susceptibility to erosion, and slow water absorption. Also, the main advantages of SLNs over liposomes rely on the higher stability and loading capacity of hydrophobic drugs [7,10]. Importantly, some of the manufacturing processes used for SLNs production can be applied to large-scale production, with excellent reproducibility being important for downstream commercial and clinical applications [11].

SLNs are the first generation of this type of lipidic particles that present some limitations, including possible polymorphic transitions, particle growth and gelling, limited cargo (although higher than liposomes), and stability of some drugs since, during storage, lipid crystallization leads to drug expulsion and premature release [10,12]. Also, like other nanoparticle-based drug delivery systems, SLNs also present high manufacturing costs compared to conventional formulations. In order to overcome the limitations of SLNs, a second-generation, nanostructured lipid carrier (NLC) was developed. NLCs differentiate from SLNs by having a blend of solid and liquid (oils) lipids in the composition. This allows a reduction in the melting point of the lipidic core and a higher loading capacity and storage stability of the system [13]. Moreover, in the last years, many lipidic nanoparticles (LNPs), some of them with mixed/hybrid properties, have been proposed, and their categorization is not always consensual. For example, Moderna’s and BioNTech/Pfizer’s COVID-19 vaccines are referred to as LNPs or SLNs, depending on the source of information [14,15]. This makes difficult the proper identification and classification of the formulations. Despite the similitudes among the SLNs, LNPs, and NLCs, in this review, we will focus on the use of classic SLNs for cancer treatment.

## 3. SLNs Production Methods

Different methods for producing SLNs have been developed and proposed over the years [16,17]. Each of them produces particles with specific characteristics and may be applied to different drugs. Thus, the production method must be chosen based on the available equipment, the properties of the drugs to be encapsulated, and the desired application. For example, methods that comprise a step at high temperatures cannot be used to formulate thermosensitive drugs. In the same way, drugs sensitive to organic solvents or sonication cannot be incorporated into SLNs using methods that require the use of organic solvents and sonication, respectively. Also, the encapsulation of hydrophilic or hydrophobic drugs requires different methods. For example, while hydrophobic drugs can be loaded using a single emulsion step, hydrophilic drugs require a double emulsion process. Also, formulations intended for parenteral administration require a more controlled size and low polydispersity of the particles and methods able to produce sterile formulations in comparison with particles designed for oral administration. Thus, to successfully develop an SLN formulation able to be translated from the bench to the bedside, it is of utmost importance to choose the proper production method based on the components of the formulations and the desired application. A brief description of the most common ones is presented below, and an overall comparison of their advantages and disadvantages is presented in Table 2. A more detailed and schematic representation of the different methods is presented elsewhere [10,16,17]. Note that new methods and variations of the presented methods are constantly being proposed.

### 3.1. Hot High-Pressure Homogenization

The hot, high-pressure homogenization (HHPH) technique is based on high-pressure homogenization (HPH). Firstly, the lipid is melted 5–10 °C above the melting point, and the drug or active component is dissolved. Then, the lipid phase is dispersed into the aqueous phase, where the surfactant was previously dissolved at the same temperature under high-speed stirring. This pre-emulsion is then homogenized by passing the liquid at high pressures (100–2000 bars) through a narrow gap (micron dimensions), achieving very high velocity. Very high shear stress and cavitation forces disrupt the particles down to the submicron range. Finally, the obtained nanoemulsion is cooled to room temperature to allow the nanoparticles to crystalize and form the SLNs [18].

### 3.2. Cold High-Pressure Homogenization

Cold-high pressure homogenization (CHPH) is similar to HHPH. The lipids with the active compound are melted 5–10 °C above the melting temperature, and it is cooled quickly with liquid nitrogen or dried ice to homogeneously distribute the drug in the solid lipid matrix. Then, the lipid phase is milled to obtain microparticles of around 50–100 μm. These solid microparticles are then dispersed in the aqueous phase at a cold temperature. Finally, as in the HHPH, the pre-suspension is homogenized at high-pressure to obtain SLNs in the range of 50–100 nm [19,20].

### 3.3. Microemulsion

This technique is based on the dilution of microemulsions. Microemulsions are thermodynamically stable, optically isotropic, and transparent. Firstly, the hydrophobic drug is dissolved in the warm lipid (55–85 °C) with the surfactant. At the same time, the water containing the co-surfactant is heated at the same temperature and dispersed into the lipid mixture. Then, the warm microemulsion is dispersed into cold water (2–3 °C) at a ratio varying from 1:25–1:50, under mechanical stirring. The SLNs dispersion is often purified by tangential ultrafiltration to remove the remaining surfactant and co-surfactant [6].

### 3.4. Double Microemulsion

The double microemulsion is similar to the microemulsion and is widely used to entrap hydrophilic drugs into the SLN. The hydrophilic drug is dissolved in water and dispersed in the melted lipid (55–85 °C), forming a water-in-oil dispersion. Then, the first microemulsion is dispersed in warm water with the surfactants and co-surfactants (as microemulsions), followed by the dispersion in cold water [5,21].

### 3.5. High Shear Homogenization

High shear homogenization (HSH) is a technique, like the ones referred to previously, performed in the absence of organic solvent. Briefly, the lipid is melted 5–10 °C above the melting point, and the drug is dissolved. The aqueous phase, with the surfactant, is heated at the same temperature and homogenized with the lipid phase by a rotor-stator homogenizer. The nanoemulsion is let to cool down to obtain the SLNs. In HSH, the amount of surfactant, the stirring, and the cooling time are very important to optimize the particle size [18].

An alternative to HSH is high-speed sonication (HSS). It has the same protocol as the HSH, but the formulation is sonicated instead of using a rotor-stator homogenizer. However, this change also presents some disadvantages, such as the possible metal contamination and damage to sensitive biomolecules [5,22].

### 3.6. Solvent Emulsification-Evaporation

The solvent emulsification-evaporation (SEE) starts with dissolving the drug and the lipid matrix in an organic solvent. Using a high-speed homogenizer, this solution is then emulsified with water containing the surfactant. Finally, the emulsion is passed through the high-pressure homogenizer to obtain the nanoemulsion and kept overnight in stirring to eliminate the organic solvent. The SLN is formed when the lipid precipitates in the aqueous solution [23].

### 3.7. Solvent Emulsification-Diffusion

The solvent emulsification-diffusion method (SED) consists of saturating a partially water-soluble solvent such as benzyl alcohol or butyl lactate with water to obtain a thermodynamic equilibrium between both liquids. The lipid is dissolved in the organic phase (water-saturated solvent) and emulsified with the aqueous phase (solvent-saturated water) containing the surfactant. Then, the emulsion is quickly diluted in water and stirred to extract the solvent into the external water phase by diffusion, thus, precipitating the lipid and generating the SLNs [24].

### 3.8. Solvent Injection

The solvent injection (SI) is a modification of the SED technique. As in SED, the lipid and drug are dissolved in a partially water-soluble solvent or a mixture of solvents. Then, the organic phase is rapidly injected through a needle into the aqueous phase containing the surfactant in constant stirring. The SLNs are formed when the solvent diffuses to the external water phase and precipitates [25].

### 3.9. Solvent Injection Lyophilization

The solvent injection lyophilization (SIL) is a modification of the SI. The procedure is the same, but instead of an aqueous phase containing surfactant, it contains lyoprotectants. Then, the system is lyophilized to obtain the dry SLNs, that, once rehydrated, forms a dispersion [26].

### 3.10. Coacervation

The coacervation method is also known as fatty acid coacervation due to the use of fatty acid alkaline salts. Firstly, a solution with a polymeric stabilizer is prepared. Then, the fatty acid sodium salt is dispersed homogenously and heated under stirring above its Kraft point. The drug is then dissolved in an organic solvent added to the solution and stirred until a unique phase, and a clear solution is obtained. Finally, an acidic solution is added drop by drop to form a suspension with a pH of around 4. The suspension is then cooled down in a water bath under stirring at a low temperature (15 °C), and the SLNs are formed [27].

### 3.11. Microwave-Assisted Microemulsion Technique

The microwave-assisted microemulsion technique (MAMT) is characterized by single-pot production. The drug, lipids, and surfactants are heated in a highly temperature-controlled microwave above the lipid melting point. The melted solution is stirred in order to obtain a hot microemulsion that is finally dispersed in cold water (2–4 °C), originating the SLNs [28].

One of the main difficulties in developing nanomedicines relies on the production scale-up to an industrial level. To enter the market, nanomedicines must be produced with reproducibility from batch-to-batch the more affordable way possible at a large scale by techniques that comply with the regulatory authorities [29]. High-pressure homogenization (HPH) is already qualified for parenteral nutrition applications, thus having scale-up potential for SLN production [16]. Also, some publications have published scale-up of SLNs production using methods such as microfluidic [30] or supercritical fluid [31]. Moreover, the SLNs that are under clinical evaluation are being produced at a large scale. Additionally, considering that various SLN formulations are used in the cosmetic field, the step of scale-up is already performed and only must be adapted for regulatory qualification, which may speed-up their entrance into the pharmaceutical market.

## 4. Solid Lipid Nanoparticles Administration Routes

One of the major benefits of the SLNs is that by manipulating their composition and physicochemical features, they can be adapted to numerous applications and administration routes. Despite being highly applied for dermal applications, SLNs have the potential to be used in other clinical applications, which is the reason why different groups have been exploring many administration routes. In fact, the majority of the works/publications regarding SLNs in recent years have been intended for parenteral administration [32].

### 4.1. Topical and Dermal Administration

SLNs are widely used for the dermal administration and treatment of skin disorders. In fact, SLNs were first used in the cosmetic field prior to dermopharmacy/medicine. Topical administration usually presents some limitations related to the poor skin penetration of drugs or skin irritation caused by the drug. SLN-based formulations, generally consisting of SLNs incorporated into creams or gels, form a thin film over the skin that provides the required hydration to the stratum corneum and enables penetration of small-size particles through the epidermis. Moreover, the presence of SLNs increases the transport of active compounds through the skin by improving drug solubilization in the formulation, drug partitioning into the skin, and fluidizing skin lipids; it also protects the active compound from light, oxidation, and hydrolysis and reduces the possible irritation caused by the direct contact of drugs with skin [4,5,12]. Therefore, several SLN formulations have been successfully developed to load glucocorticoids, retinoids, anti-inflammatories, and antimycotics, among other drugs [33,34].

Interestingly, due to their solid lipid core, SLNs present the capacity to reflect ultraviolet (UV) radiation. Therefore they have been widely applied in producing several sunscreens to improve UV protection [33]. SLNs also give a smoother texture to the cream compared to conventional cosmetics [35].

### 4.2. Pulmonary Administration

The pulmonary administration route allows a non-invasive local and systemic drug administration. Contrary to the oral route, the administered drug is absorbed in the alveolar epithelium with a large absorption area, highly permeable alveolar epithelial membrane, and high vascularization with limited hepatic first-pass metabolism. In this type of administration, the formulation should be nebulized through an inhaler device. For that, all of the formulation’s physicochemical features should be controlled to obtain nebulized particles with the perfect size [12,36,37].

There are several studies where SLNs are administered through the pulmonary route for the treatment of diverse pulmonary pathologies, namely tuberculosis [38,39], asthma, chronic obstructive pulmonary disease (COPD) [40], and lung cancer [41,42,43]. Furthermore, there are SLN treatments administered through the pulmonary route for systemic diseases such as diabetes [44] and hypertension [45].

### 4.3. Oral Administration

The oral route is non-invasive, not requiring medical assistance for drug administration. Therefore, is the preferred route of administration by patients and the most desired in clinical practice. Several authors have studied the stability of SLNs in dried or suspension form for oral administration [4,12]. There are some examples of SLNs orally administered in order to improve the bioavailability of the loaded drugs; this is the case of lopinavir (for HIV treatment) [46], risperidone (schizophrenia treatment) [47], rifampicin (tuberculosis) [48], and hydrochlorothiazide (pediatric hypertension) [49].

### 4.4. Ocular Administration

Ocular delivery is a complex route of administration due to the sensitive inner structure of the eye and the poor penetration of drugs in this tissue. Therefore, the drugs are usually locally administrated using eyedrops. In this type of administration, SLNs benefit from their small size, meaning they do not obstruct vision, which happens with other formulations. Additionally, SLNs can reduce the clearance by the eye’s protective mechanisms due to adhesive properties related to nanometric size. Some formulations have been tested in animal models for treating diverse ocular diseases, like retinitis pigmentosa [50], glaucoma [51], and ocular bacterial infections [52].

### 4.5. Intravenous Administration

Intravenous administration is the most common administration route for chemotherapeutic drugs. Nanomedicine has contributed to increasing the stability and bioavailability of drugs when intravenously administrated. The formulations should be deeply characterized in order to be in accordance with the requirements for this administration route, mainly the small size, serum stability, and hemocompatible components. Accordingly, to their compositions, SLNs are perfectly able to fulfill all the requirements for adequate intravenous administration. The majority of the developed intravenous SLNs are intended for cancer therapy (deeply discussed in the next sections of the present review). However, there are also studies using intravenously administered SLNs for neurodegenerative diseases [53] as well as for the administration of contrast agents for imaging and diagnosis [54,55].

### 4.6. Intranasal Administration

Managing central nervous system (CNS) disorders is challenging due to the need for drugs to cross the blood-brain barrier (BBB) and reach the brain. The lipid nature of lipid-based nanocarriers such as SLNs can facilitate the transition across the BBB and translocate them into the brain through passive diffusion [56]. Few studies using SLNs have been proposed for the nose-to-brain delivery of drugs [57]. Although the exact mechanism of drug transport from the nose to the brain is not fully understood, and its effectiveness in humans is unclear, intranasally administered SLNs have been shown to be more effective in crossing the BBB than other formulations and administration routes.

## 5. SLNs’ Advantages for Cancer Treatment

The higher specificity of nanoDDS to the tumor site is commonly explained by the well-known permeation and retention effect (EPR effect) (Figure 2A). Accordingly, due to their nanometric size, nanoDDS can profit from the weak vessel structure and the small fenestras (300–900 nm) present in the tumor vasculature while suppressed lymphatic drainage causes retention within the tissue, thus easily reaching the tumor site by passive targeting. After tissue accumulation, nanoDDS are more prone to suffer cell internalization. The mechanism for SLN internalization into cells usually occurs through clathrin- or caveolin-mediated endocytosis (Figure 2B). Briefly, the plasmatic membrane surrounds the SLNs, forming a vesicle. This vesicle will suffer different stages of maturation, beginning with the early endosome and finishing in the lysosome. Depending on the SLN’s superficial charge and composition, it can be degraded in the endosome or lysosome, releasing its content into the cytosol [58]. Moreover, by modulating the composition of the nanoDDS, they can be easily functionalized with different targeting moieties to favor active targeting against the type of cell of interest, which allows a highly specific treatment compared with the usual unspecific conventional therapy.

Regarding SLN formulations under clinical evaluation, when searching for lipid nanoparticles, despite the difficulties in identifying the type of formulation used, some lipidic nanoparticles, including SLNs, appear as enrolled in clinical trials for the treatment of different diseases. The majority are intended for topical [59,60] and oral [61] administration; however, to our knowledge, no clinical trials are ongoing to study the use of SLNs for cancer therapy. Despite the absence of an actual clinical evaluation, based on the number of works published and formulations under development for cancer treatment, it is expected the translation of some of them from the bench to clinical trials in the near future. In fact, boosted by the success of COVID-19 vaccines, the interest in lipidic nanoparticles exponentially increased, and the pharmaceutical economic studies from Fortune Business Insights predict an increase in the global market share of lipidic nanoparticles, including SLNs [62].

As referred to, SLN development began in the 1990s, with the first formulations reaching the market in the cosmetic field. With the increasing interest in this type of system, many patents related to the composition and techniques used have been presented. An example is an SLN formulation to improve sorafenib bioavailability [63]. Also, another patent relates to silymarin-loaded SLNs targeting tumor cells through folic acid surface modification [64]. The authors claimed an active lung tumor targeting effect, with improved bioavailability and reduced toxic effects. Oral delivery of docetaxel by means of SLNs was also claimed in a patent for the improvement of solubility, and sustained release effect of the drug [65]. An extended list of patents is reviewed elsewhere [66].

## 6. SLNs for Drug Delivery

Over the years, several researchers have developed SLN-based formulations for the encapsulation of different anti-cancer drugs (Table 3). The ultimate goal of all these studies was to reach an efficacious and non-toxic formulation that could be used as a clinical alternative to improve the therapeutic index of conventional drugs. Due to the high number of reported studies, we will focus on some of the most recent publications in this review. For example, M.C. Leiva et al. [67] used glyceryl tripalmitate SLNs loaded with paclitaxel (PTX) for breast and lung cancer treatment. They observed that the encapsulated PTX was more effective in inhibiting cell proliferation than the free drug, thus improving the drug’s therapeutic index. Another example of drug efficacy improvement was demonstrated by N. Clemente et al. encapsulating temozolomide (TMZ) for melanoma treatment. Their SLN-TMZ was more efficient than free TMZ in cell cultures, and in vivo, they reduced the tumor growth and increased mouse survival by administrating a lower dose of the drug [68].

One of the major challenges regarding cancer therapy relies on the development of drug resistance by cancer cells. Usually, after several cycles of chemotherapy, cancer cells start to activate multi-drug resistance (MDR) mechanisms [2,69]. Most commonly, MDR is caused by the overexpression of drug efflux pumps, such as the P-glycoprotein (P-gp), which utilizes ATP-derived energy to pump chemotherapy drugs out of tumor cells and protect tumor tissues from chemical toxicity [70,71]. This problem was studied by G. Guney Eskiler et al. [72] and B. Stella et al. [73], who were able to avoid drug resistance using SLNs. In the work of G. Guney Eskiler et al. [72], stearic acid-based SLNs encapsulating tamoxifen (TMX) were able to reduce the cell viability of TMX-resistant breast cancer cell line (MCF7-TamR). In the case of B. Stella et al. [73], a sodium behenate SLN encapsulating squalenoyl doxorubicin (SQ-Dox) increased the inhibition of Dox-resistant ovarian cancer (A2780-DoxR) cell growth and the capacity of colony formation when compared with the cells treated with the free drugs. In another study, to overcome MDR in breast cancer, Wenrui Wang et al. [71] studied the in vitro and in vivo efficacy of a resveratrol (Res)-loaded SLNs formulation with and without D-α-Tocopheryl polyethylene glycol 1000 succinate (TPGS), a derivative of natural vitamin E (α-tocopherol) that inhibits the activity of ATP-dependent P-gp. It was found that SKBR3/PR cells treated with TPGS containing SLNs (TPGS-Res-SLNs) exhibited significant inhibition of cell migration and invasion, as compared with free Res and SLNs without TPGS (Res-SLNs). In addition, TPGS-Res-SLNs promoted more apoptosis of tumor cells and induced higher tumor reduction in SKBR3/PR xenografts that the free Res or the Res-SLNs, owning a better therapeutic outcome (Figure 3).

Once the encapsulation of a single drug in an SLN with a certain composition becomes trivial, several authors innovate by creating stimuli-responsive formulations or systems for dual therapy, diagnostic, or even theragnostic. For example, A. Grillone et al. [74] and T. Liu et al. [75] produced SLNs with superparamagnetic iron oxide nanoparticles (SPIONs) and Nutlin-3 or Dox, respectively. A. Grillone et al. [74] successfully delivered the nanoformulation through the BBB for the treatment of glioblastoma. T. Liu et al. [75] developed a pH-sensitive SLN with Dox and SPIONs to reduce tumor growth more efficiently due to the combination of magnetic nanoparticle-driven thermal therapy and chemotherapy in vivo.

The use of stimuli-responsive lipids can also be a crucial strategy in the context of controlled and sustained drug release since it allows an accurate manipulation of the drug release profile. For example, through the thermal properties of the lipids, it is possible to increase the drug release at a specific body region, as demonstrated by M. Rehman et al. [76]. They developed a thermosensitive SLN (mixture of lauric acid with oleic or linoleic acid) loaded with 5-fluoracil (5-FU) for breast cancer. This SLN releases quickly and higher doses of 5-FU at high temperatures, such as the tumor temperature, thus increasing the drug availability in the tumor and its therapeutic index.

Another example is the pH-sensitive SLN developed by G. Zheng et al. [77] which added adipic acid dihryzacide to the glycerin monoestereate (GMS) matrix. With this modification, the presented nanoformulation loaded with Dox had a preferential release in the tumor acidic environment (pH = 5), being possible to observe an initial burst release of the drug (within 8 h) followed by a more sustained release in time (until 96 h). This work is a representative example of the high versatility of SLNs since, by the simple modification of SLNs composition, it is possible to manipulate the release of chemotherapeutic drugs and adapt such release to the desired application.

While several authors are dedicated to improving the therapeutic index of conventional drugs, other researchers are more focused on substituting such drugs for more natural compounds or at least reducing the dose of the chemotherapeutic by combination with natural substances. This is the case of the works by J.S. Baek et al. [78] and W. Wang et al. [79], where stearic acid-based formulations were used for the encapsulation of wogonin and curcumin, respectively. These two natural substances are very active in vitro but, due to their unfavorable pharmacokinetics, require advanced delivery systems such as nanoDDS for in vivo administration. Guorgui et al. [80] also loaded curcumin in stearic acid-based SLNs and TPGS nanoparticles. They obtained higher curcumin plasma levels in mice using the SLNs and observed tumor growth regression in Hodgkin’s Lymphoma mice models. Moreover, when given in combination with bleomycin, Dox, and vinblastine, curcumin showed an additive growth inhibitory effect. The authors concluded that once adequately formulated, curcumin can be used as an adjuvant agent for the treatment of Hodgkin’s Lymphoma.

**Table 3 pharmaceutics-15-00831-t003:** Examples of SLNs formulation for cancer treatment in different stages of development.

Lipid	Method	Drug	Cancer	Phase	Ref
SA	Hot sonication	Wogonin	BC	In vitro	[78]
SA	Emulsification-solidification	Curcumin	BC	In vitro	[79]
SA & lecithin & TPGS	Emulsification-solidification	Resveratrol	BC	In vivo	[71]
CP	SEE	Nutlin3 & SPIONS	GBM	In vitro (BBB model)	[74]
Trilaurin	Microemulsion cold dilution	Curcumin	PC	In vivo	[81]
Gliceryl tripalmitate	HSH	PTX	BC and LC	In vitro	[67]
Resveratrol-stearated	Microemulsion	Omg-3	CRC	In vitro	[82]
SA	HSH	TMX	BC	In vitro	[72]
Lauric acid & (linoleic acid/oleic acid)	Hot melt encapsulation	5-FU	BC	In vitro	[76]
Sodium behenate	Coacervation	TMZ	Melanoma	In vivo	[68]
Behenic acid sodium salt	Coacervation	DOX	OC	In vitro	[73]
SA	SEE	Curcumin	HL	In vivo	[80]
Trilauriun & TPGS	Microemulsion	DOX and SPIONS	PTC (murine)	In vivo	[75]
CP	HPH	Indirubin	GBM	In vitro	[83]
CP	SEE	Topotecan hydrocloride	CC	In vitro	[84]
GMS	HSH	Talazoparib (BMN673)	BC	In vitro	[85]
Myristyl myristate	Hot sonication	Linalool	HC and LC	In vitro	[86]
GMS	Hot sonication	PTX and ascorbyl palmitate	Melanoma (murine)	In vivo	[87]
SA & lecithin	SIL	5-FU	Melanoma	In vivo	[88]
Compritol^®^	Microemulsion	AP9-cd (ligand)	Leukaemia	In vivo	[89]
1-tetradecanol	HHPH and ultrasonication	Temoporfin	BC	In vivo	[90]
GMS, AAD, and RGD	Emulsification-solidification	DOX	BC	In vivo	[77]
SA	SED	PTA	OC	In vitro	[91]
Precirol^®^ ATO5	Emulsification-solidification	PTX	BC	In vitro	[92]
Imwitor^®^ 308 and Dynasan^®^ 114	Ultrasonic melt-emulsification	Celecoxib	CRC	In vitro	[93]
Lecithin & DSPE-PEG2000	Film dispersion method	PTX, Curcumin	LC	In vivo	[94]
GMS	modified emulsion/solvent evaporation method	Abiraterone Acetate	PC	Ex vivo/In vivo	[95]

Abbreviations: Lipid: Stearic Acid (SA), DL-α-Tocopherol methoxypolyethylene glycol succinate (TPGS), Cetyl Palmitate (CP), Glycerol Monostearate (GMS), Adipic acid dihydrazide (AAD), Arginine-Glycine-Aspartic (RDG). Method: Microwave-Assisted Microemulsion Technique (MAMT), Solvent Emulsification-Evaporation (SEE), High Shear Homogenization (HSH), Solvent Injection-Lyophilization (SIL), Hot High-Pressure Homogenization (HHPH), Solvent Emulsification-Diffusion (SED). Drug: Paclitaxel (PTX), Tamoxifen (TMX), 5-fluorouracil (5-FU), Temozolomide (TMZ), Superparamagnetic iron oxide nanoparticles (SPIONs), Doxorubicin hydrochloride (DOX), 1,3,5-triaza-7-phosphaadamantane (PTA). Cancer: Breast Cancer (BC), Glioblastoma (GBM), Prostate Cancer (PC), Lung Cancer (LC), Colorectal Cancer (CRC), Ovarian Cancer (OC), Hodgkin’s Lymphoma (HL), Papillary Thyroid Cancer (PTC), Prostate Carcinoma (PC), Cervical Cancer (CC), Hepatocarcinoma (HC).

## 7. Targeted SLNs

One of the challenges of cancer therapy is the management of adverse side effects caused by conventional chemotherapy that affect not only the cancer cells but also the healthy cells. In order to solve this problem, different approaches to specifically target tumor cells have been strongly explored in recent years. NanoDDS are one of the adopted strategies since their surface can be functionalized with several targeting moieties to increase the uptake of the delivery system by the target cells. In the case of SLNs, different molecules have been used to introduce active targeting moieties, and a summary of the most recent and relevant works is shown in Table 4.

It is widely known that some cancer cells overexpress certain receptors in comparison to healthy cells; thus, by targeting these receptors, it is expected to obtain a higher uptake by cancerous cells, as demonstrated by E. Souto et al. [96] and S. Shi et al. [97]. E. Souto et al. [96] developed a Compritol^®^-based SLN with active targeting against HER2 receptor by conjugating the compact antibody CAB51. The SLNs showed higher cellular uptake in BT-474 (HER2 positive cells) cells than in MCF-7 (HER2 negative cells), proving the active targeting of the HER2 receptor. S. Shi et al. [97] proposed dual-targeting by functionalizing the SLN’s surface with hyaluronic acid and tetraiodothyoacetic acid as ligands of CD44 and αvβ3 receptors, respectively. Functionalized SLNs had a higher cellular uptake in cells expressing those receptors and increased the therapeutic effect of docetaxel in vivo. Another example is the SLNs conjugated with the antibody against the receptor for advanced glycation end products (RAGE) developed by V. T. Siddhartha et al. [98]. These SLNs loaded with di-allyl-disulfide were taken up more efficiently and induced higher cytotoxicity than the unconjugated SLNs in triple-negative breast cancer cells. The higher cytotoxicity was due to a combined effect of intracellular accumulation of di-allyl-disulfide and the promotion of pro-apoptotic proteins by the inhibition of the RAGE receptor.

Targeting the folate receptor is also a widely used approach for increasing the tumoral accumulation/internalization of SLNs, as demonstrated by R. Rosière et al. [99], J.S. Baek et al. [100], and K. Rajpoot et al. [101], among others. In the case of R. Rosière et al. [100], they developed an SLN coated with a folate-grafted copolymer of PEG and chitosan for lung tumors. They observed that this nanoformulation loaded with PTX had a higher penetrating capacity into lung tumors when administrated by inhalation rather than the conventional treatment. Due to folate functionalization, the SLN does not have to rely on tumor vascularization to reach the tumor site. K. Rajpoot et al. [101] encapsulated oxaliplatin into SLNs conjugated with folic acid for colorectal cancer treatment and observed that the SLNs functionalized had a higher cytotoxicity effect in the HT29 cell line compared to the unfunctionalized SLNs. They believed this higher cytotoxicity was due to the higher cellular uptake of the nanoformulation mediated by the folate receptor. SLNs are also useful for transporting plant essential oils (fatty nature) due to their ability to encapsulate hydrophobic compounds. Based on this, S.F. Tabatabaeain et al. [102] successfully loaded *Satureja khuzistanica* essential oil into SLNs modified using a chitosan coating attached to folic acid to improve drug accumulation into breast cancer cells (MCF-7 cells). J.S. Baek et al. [100] developed SLNs coated with folic acid and conjugated to stearic acid for the treatment of MDR breast cancer. These SLNs loaded with curcumin and PTX showed a higher cellular internalization and synergic cytotoxicity partially due to the capacity of curcumin to inhibit P-gp, thus increasing the PTX intracellular accumulation.

The use of sugars has also been widely studied for targeting cancer cells due to their higher energetic metabolism. For that, N. Soni et al. [103] and N.K. Garg et al. [104] functionalized their SLN’s surface with mannose and fucose. N. Soni et al. [103] loaded gemcitabine in a stearyl amide-based SLNs for lung cancer. Adding mannose to the SLN’s surface increased the cellular uptake through the mannose receptors overexpressed in the macrophages. Also, in biodistribution studies, the SLNs functionalized showed more accumulation in the lungs than the basal SLNs. N. K. Garg et al. [104] used fucose-coated SLNs loaded with methotrexate for breast cancer treatment, achieving approximately 70% of tumor reduction.

Several treatments and targeting actives can be combined in the same SLNs. M.Y. Shen et al. [105] developed SLNs loaded with Dox and SPIONs with a double coating of folic acid and dextran for colon cancer treatment. The folic acid increased the cellular uptake through the folate receptors, and the dextran increased the tissue-specificity of SLNs since the enzyme capable of degrading dextran is only present in the colon. Besides, the coating also allowed the oral administration of SLNs by protecting them from degradation at harsh gastric conditions. This nanoformulation showed promising results in vitro and in vivo, not only by the high tumor cytotoxicity but by the lack of side effects. This is an example of a multifunctional nanoparticle that is expected to be the future of nanomedicine by combining different components and functions in the same vehicle for improved biological behavior and clinical outcome.

Another widely used approach to improve cancer treatment is the functionalization of nanoDDS with cell-penetrating peptides (CPP), as reported by B. Liu et al. [106]. They have modified the SLNs surface with a trans-activating transcriptional activator (TAT) peptide and studied how the functionalization can increase the accumulation of SLNs loaded with PTX and α-tocopherol succinate-cisplatin prodrug. They report a higher accumulation of the SLNs in the tumorous cells, increasing the drugs’ therapeutic index. They discuss that this higher accumulation is due to the presence of TAT. However, since the function of CPP is to translocate through the plasmatic membrane, not only to the tumor cells but also the healthy cells, the use of CPP should not be considered active targeting, contrary to what has been described in some literature. The use of CPP as moieties for nanoDDS surface functionalization relies on the improvement of cellular uptake and consequent improvement of drug effectiveness without tumor specificity. Even without this cell-specificity, the use of CPP is of major interest in brain tumors that require crossing the BBB. The use of SLNs modified with moieties that increase the permeability of drugs through the BBB has been proposed by Y.C. Kuo et al. [107,108] and A. Kadari et al. [109], among others. Y.C. Kuo et al. [107,108] developed different Compritol^®^-based SLNs formulations for the encapsulation of etoposide as a chemotherapeutic drug. One of the formulations was composed of a double surface conjugation with an 83–14 monoclonal antibody (83–14 MAb) and the anti-epithelial growth factor receptor (AEGFR) [107]. The double targeting facilitated the BBB crossing due to the recognition of the α-subunit insulin receptor by the 83–14 MAb, and the active targeting of cancerous cells was achieved by the AEGFR. Another formulation developed by the same authors included an anti-melanotransferrin antibody attached to the surface of SLNs to increase the BBB crossing [108]. Both studies were tested in an in vitro model of BBB and glioblastoma and successfully decreased the glioblastoma cells proliferation without causing cytotoxicity to the BBB. A. Kadari et al. [109] have developed an SLN loaded with docetaxel with an angiopep-2 linked to its surface. Angiopep-2 is a specific ligand for lipoprotein receptor-related protein 1 (LRP1), a receptor overexpressed in the BBB and the glioma cells. The in vivo results demonstrated a higher accumulation of the SLNs in the brain and an increase in the survival rate of the animals compared with the current treatment, thus arising as a promising alternative for the treatment of glioblastoma.

**Table 4 pharmaceutics-15-00831-t004:** Summary of targeted SLNs for different cancer types in different stages of development. N.A.—not applicable.

Lipid	Method	Drug	Targeting	Cancer	Phase	Ref
Compritol^®^	HSH	N.A.	Antibody against HR2	BC	In vitro	[96]
Glyceryl stearate & Chol	Nanoprecipitation	PTX	Folic acid coated	LC	In vivo	[99]
SA and lecithin	HHPH	*Satureja khuzistanica* Essential Oil & folate-bound chitosan	Folic acid	BC	In vitro	[102]
Behenic acid	Coacervation	Methotrexate	Apoemimkin Chimera	GBM	In vivo	[110]
GMS and SA	SEE	Docetaxel	Angiopep-2	GBM	In vivo	[109]
PA	SED	di-allyl-disulfide	RAGE antibody	BC	In vitro	[98]
DSPE	SED	Oxaliplatin	Folic acid	CRC	In vitro	[101]
GMS and TPGS	SEE	PTX and Curcumin	SA-folate	BC	In vitro	[100]
GMS, SPC, and Oda	Film-ultrasonic method	Docetaxel	HA-Te	BC	In vivo	[97]
Tristearin	SED	Irinotecan	Folic acid	CRC	In vivo	[111]
GMS	SEE	PTX and TSC	TAT	CC	In vivo	[106]
Trilaurin and TPGS	Microemulsion	DOX and SPIONs	Folic acid	CRC	In vivo	[105]
Glyceryl palmitostearate	Emulsification-solidification	PTX	Anti CD44v6 antibody	BC	In vitro	[112]
Stearyl amine	SEE	Gemcitabine	Mannose	LC	In vivo	[103]
Gelucire^®^ 50/13	Microemulsion	Methotrexate	Fucose	BC	In vivo	[104]
GMS	HHPH	PTX	Hyaluronic acid	CC & BC	In vivo	[113]
SA	SI (modified)	Resveratrol and Ferulic acid	Folic acid	CRC	In vitro	[114]
SA	Hot melted sonication	PTX	HP-β-CD	BC	In vivo	[115]
GMS and Compritol^®^	Hot melt-emulsification	Dox and Curcumin	Folic acid	BC	In vivo	[116]
Compritol^®^, tripalmitin, SA, and Chol	SEE	ETP	Melanotransferr-in antibody and Tamoxifen	GBM	In vitro	[108]
GMS	SEE	PTX	Wheat germ agglutinin	LC	In vivo	[117]
Compritol^®^, CL, and SA	Microemulsion	ETP	83–14 MAb and AEGFR	GBM	In vitro	[107]
Behenic acid, tripalmitin, and cacao butter	SEE	Carmustine	Tamoxifen and Lectoferrin	GBM	In vitro	[118]
Tristearin & HSPC	SI	PTX	Lactoferrin	LC	In vitro	[39]
PA and Dynasan^®^	N.A.	Saquinavir	83–14 MAb	GBM	In vitro	[119]
Compritol^®^ 888 ATO and Precirol^®^ ATO	Sonication of pre-emulsion	DTX	AS1411 anti-nucleolin aptamers	CRC	In vivo	[120]
Cetyl palmitate	Hot ultrasonication method	Mitoxantrone	Folate receptor	BC	In vitro	[121]

Abbreviations: Lipid: Stearic Acid (SA), Glycerol Monostearate (GMS), Palmitic acid (PA), 1,2-distearoyl-sn-glycero-3-phosphoethanolamine (DSPE), DL-α-Tocopherol methoxypolyethylene glycol succinate (TPGS), Soy phosphatidylcholine (SPC), Octadecylamine (Oda), Cardiolipin (CL), Hydrogenated soya phosphatidylcholine (HSPC). Method: High Shear Homogenization (HSH), Hot High-Pressure Homogenization (HHPH), Solvent Emulsification-Evaporation (SEE), Solvent Emulsification-Diffusion (SED), Solvent Injection (SI). Drug: Paclitaxel (PTX), α-tocopherol succinate-cisplatin (TSC), Doxorubicin hydrochloride (DOX), Superparamagnetic iron oxide nanoparticles (SPIONs), Etoposide (ETP), Docetaxel (DTX). Cancer: Breast Cancer (BC), Lung Cancer (LC), Glioblastoma (GBM), Colorectal Cancer (CRC), Cervical Cancer (CC), Prostate Cancer (PC), Ovarian Cancer (OC).

## 8. SLNs for Gene Delivery

Gene therapy gained a huge interest in the field of cancer treatment to overcome the high toxicity and the low specificity associated with conventional drugs. Gene therapy allows the expression of an exogenous oligonucleotide, encoding for a missing or defective gene or achieving the silence of a particular gene [122,123]. The exogenous expression is usually mediated by plasmid DNA (pDNA) which consists of a small circular sequence of DNA that can replicate after entering the nucleus [124]. Gene silencing, on the other hand, is usually performed by technologies of RNA interference (RNAi). There are three types of RNAi: short hairpin RNA (shRNA), small interfering RNA (siRNA), and microRNA (miRNA). The shRNA, like the pDNA, has its function in the nucleus, where it is transcribed to small RNA and binds to the complementary mRNA, thus inhibiting the expression of the protein [125]. On the other hand, siRNA and miRNA do not have to enter the nucleus since they exert activity in the cytoplasm [126]. In the case of the siRNA, it is a 20–25 base nucleotide with a specific sequence that targets and degrades specific mRNAs [127]. Regarding miRNA is not as specific as siRNA since the same miRNA can bind to different mRNAs [128,129].

The major issue regarding gene therapy is the delivery of genetic material into the cells. The oligonucleotide’s physicochemical properties make its entrance into the cell difficult without a proper vector. Despite their low packaging capacity, high production costs, and immunogenicity, viral vectors are still the preferred vectors for gene delivery in the clinical setting. In order to overcome all the drawbacks associated with the viral vectors, considerable investment and efforts have been made to develop non-viral vectors [130,131], including some of the recent vaccines against the SARS-CoV-2 virus [14]. In this review, we are focused on the application of SLNs for gene therapy in oncological diseases.

Many formulations have been proposed with differences regarding the components and the physicochemical properties of the SLNs enjoying the lipidic nature of the particles to promote cell internalization. However, many of them have at least one cationic lipid and/or phospholipid, such as 1,2-dioleoyl-3-trimethylammonium propane (DOTAP), in the composition to facilitate the complexation of genetic material. Table 5 summarizes different SLNs developed for cancer gene therapy, where the main components used for its formulation can be observed. J. Jin et al. [132] developed an SLN complexed with c-Met siRNA for glioblastoma. The c-Met complexation with SLNs increased its accumulation in the brains of mice, leading to a decrease in tumor growth. C. Botto et al. [133] have successfully loaded the shNUPR1 in a Precirol^®^-based SNL and delivery it to the cells. Moreover, SLNs/shNUPR1 downregulated the NUPR1 gene, which is known to cause chemoresistance and cancer proliferation in hepatocellular carcinoma. Several works describe the use of SLNs to downregulate the expression of signal transducer and activator of transcription 3 (STAT 3). M. Kotmakçi et al. [134] used shRNA against STAT3 to reduce STAT3 levels and re-sensitize resistant lung cancer cells (CR-Calu1) to cisplatin. Zhang et al. used decoy oligodeoxynucleotides (ODN) to target STAT-3. The results were promising, showing an inhibition of tumor growth activating the apoptotic cascade, regulating autophagy, and reversing the epithelial-mesenchymal transition program with no obvious toxicity on nude mice [135].

In many studies, gene therapy is combined with chemotherapeutic drugs within the same nanoparticle to potentiate their synergy, as reported by G. Büyükköroglu [136], Y.H. Yu et al. [137] and T. Li et al. [138], among many others. In the case of G. Büyükköroglu [136], they loaded SLNs with Bcl-2 siRNA and PTX for the treatment of cervical cancer. Their results show higher cytotoxicity with the combination of siRNA and PTX than with individual treatments. Also, they propose a local administration by encapsulating the SLNs in a PEG suppository to reduce the systemic exposure of the pharmacologic compounds and thus reduce their side effects [139]. Y.H. Yu et al. [137] developed cationic SLNs for breast cancer treatment loaded with PTX and MCL1-siRNA that successfully reduced tumor growth in vivo. T. Li et al. [138] studied the synergetic anticancer activity of sorafenib and all-trans retinoic acid (ATRA) combined with miRNA-542-3p loaded in SLNs. Again, the SLNs showed a much higher cell growth inhibition than individual treatments, such as free drugs or in SLNs. Moreover, in vivo, it also showed promising results by an increased reduction of tumor growth and increased blood circulation time compared to the free drugs. They believe this high antitumor efficacy is due to the combined therapeutic effects of drugs with the microRNA.

Targeted SLNs have also been developed for gene delivery. D.M. Yu 2016 et al. [140] loaded SLNs with PTX and pDNA and attached hyaluronic acid (HA) with a pH-sensitive linker to promote release at low pH. This nanoformulation not only efficiently released the chemotherapeutic drug and the pDNA in breast cancer cells but also increased the accumulation and preferential release of the compounds in the tumor tissue by actively targeting HA to CD44 receptors.

Finally, different nanoDDS can be successfully combined for improved therapy, as demonstrated by V. Juang et al. [141]. They developed an SLN and a liposome loaded with miRNA200 and irinotecan for colon cancer treatment. Both nanoformulations had their surface modified with three different peptides (including one cell-penetrating peptide, one peptide targeting tumor neovasculature undergoing angiogenesis, and one mitochondria-targeting peptide) that increased the cellular uptake and provided an active target to tumoral cells. Moreover, this nanoformulation was coated with a pH-sensitive PEG-lipid that enhanced the release of the miRNA in tumor sites due to the acidic environment. The combination therapies of different delivery systems promoted the synergetic activity between irinotecan and miR-200 since miR-200 increased the cancer cell sensitivity to irinotecan. The in vivo studies showed a highly effective inhibition of tumor growth caused by suppressing several proteins such as Rac-1, KRAS, and β-catenin, among others, being a promising study for further development.

**Table 5 pharmaceutics-15-00831-t005:** SLNs with different genetic material for different cancer types in different stages of development. N.A.—not applicable.

Lipid	Method	Drug	Genetic Material	Cancer	Phase	Ref
Gelucire^®^ 50/13	SEE	PTX	Bcl-2 siRNA	CC	In vitro	[136]
Precirol ATO5 and Compritol^®^	Microemulsion (modified)	N.A.	pDNA stat3	LC	In vitro	[134]
Steryalmide	Microemulsion	Sorafenib/ATRA	miR-542-3p	GC	In vivo	[138]
DSPE, DOTAP, and αPC	SED	Irinotecan	miR200	CRC	In vivo	[141]
DOPE, Chol, and DC-Chol	SEE	N.A.	c-Met siRNA	GBM	In vivo	[132]
DOPC	emulsification solidification methods	PTX	MCL-1 siRNA	BC	In vivo	[137]
GMS	film-ultrasonic dispersion method	PTX	pDNA	BC	In vivo	[140]
Precirol ATO5	ethanolic precipitation technique & HPH	N.A.	shNUPR1	HC	In vitro	[133]
DOTAP and GMS	SED	N.A.	miR200	BC	In vitro	[142]
GMS, SPC, and Chol	film-ultrasonic method	N.A.	miR-34a	LC	In vivo	[143]
GMS and SPC	SED	N.A.	AMO	LC	In vitro	[144]
GMS and soya lecithin	solvent diffusion method	N.A.	STAT3 decoy oligodeoxynucleotides	OC	In vivo	[135]
Cetyl palmitate and Cremephor RH 40, Peceol, and propylene glycol	melt-emulsification technique	N.A.	siRNA-EGFR siRNA-PD-L1	GBM	In vivo	[145]

Abbreviations: Lipid: 3β-[N-(N′,N′-dimethylaminoethane)-carbamoyl]-cholesterol (DC-Chol), 1,2-distearoyl-sn-glycero-3-phosphoethanolamine (DSPE), 1,2-Dioleoyl-3-trimethylammonium propane (DOTAP), l-α-Phosphatidylcholine (αPC), 1,2-Dioleoyl-sn-glycero-3-phosphocholine (DOPC), Glycerol Monostearate (GMS), Soy phosphatidylcholine (SPC). Stearic Acid (SA), Glycerol Monostearate (GMS), Palmitic acid (PA), DL-α-Tocopherol methoxypolyethylene glycol succinate (TPGS), Soy phosphatidylcholine (SPC), Octadecylamine (Oda), Cardiolipin (CL), Hydrogenated soya phosphatidylcholine (HSPC). Method: High Shear Homogenization (HSH), Hot High-Pressure Homogenization (HHPH), Solvent Emulsification-Evaporation (SEE), Solvent Emulsification-Diffusion (SED), Solvent Injection (SI). Drug: Paclitaxel (PTX), α-tocopherol succinate-cisplatin (TSC), Doxorubicin hydrochloride (DOX), Superparamagnetic iron oxide nanoparticles (SPIONs), Etoposide (ETP). Cancer: Cervical Cancer (CC), Lung Cancer (LC), Gastric Cancer (GC), Colorectal Cancer (CRC), Glioblastoma (GBM), Breast Cancer (BC), Hepatocarcinoma (HC), Ovarian Cancer (OC).

## 9. Other Applications

### 9.1. Immunotherapy

The idea of using a patient’s immune system to fight cancer lead to the concept of cancer immunotherapy. In the last years, this approach increased exponentially, becoming an important and promising alternative at the clinical level.

Cancer cells have the capacity to make themselves invisible to the immune system by selecting for certain genetic changes, by having proteins on their surface that turn off immune cells (e.g., PD-L1), or by inducing changes in the surrounding stroma. Different approaches have been proposed in this scenario to activate the immune response. This includes monoclonal antibodies, immune checkpoint inhibitors, adaptive T-cell transfer, and cancer vaccination [146]. Specifically, here are limited studies in the literature related to cancer immunotherapy with SLNs.

G. Erel-Akbaba et al. [145] developed an iRGD (CCRGDKGPDC)-conjugated SLN to deliver siRNA against both epidermal growth factor receptor (EGFR) and PD-L1 for glioblastoma treatment. Moreover, 5 Gy radiation pre-treatment led to enhanced transportation of SLNs to glioblastomas, yielding activation of an immune response, slower tumor growth, and prolonged mouse survival (Figure 4).

### 9.2. Imaging

The advances in imaging techniques acquired major importance in the cancer field since they allow the achievement of an accurate early diagnosis and monitoring of the patient during and after the treatment. This can substantially impact the treatment efficacy and patient survival [147]. There are multiple techniques for imaging, and some of them need contrast agents for better differentiation between the tissues. The contrast agents are administered intravenously, being one of their major issues the lack of specificity to the site of interest. J. Sun et al. [55] have developed SLNs loaded with gadolinium diethylenetriaminepentaacetic acid (Gd-DTPA), a widely used contrast agent in magnetic resonance imaging (MRI), for colon cancer imaging. Due to the encapsulation of the Gd-DTPA, they have been able to increase its cellular uptake and the tumor’s resolution and differentiation, and limits. Also, an in vivo biodistribution of SLNs-Gd-DTPA showed a high accumulation in the tumor site.

### 9.3. Theragnostic

Theragnostic consists of the combination of treatment and diagnosis in the same system. Cancer treatment is a promising new strategy that will contribute not only to early tumor diagnosis but also allow constant monitoring of the treatment’s effectiveness [148,149]. J. Kulbacka et al. [150] and Y. Kuang et al. [151] developed SLNs for cancer theragnostics. J. Kulbacka et al. [150] developed cetyl palmitate-based SLNs loaded with cyanine IR-780 and flavonoid derivates for colon cancer. The flavonoid was used as the drug in this nanoformulation and the cyanine IR-780 as a diagnostic agent and a photosensitizer. The encapsulation of cyanine IR-780 decreased its cytotoxicity and increased its efficacy. Y. Kuang et al. [151] also loaded IR-780 in SLNs for phototherapy and imaging. This nanoformulation enabled the specific treatment due to the guided imaging that allows light activation in the area of interest. Finally, K.H. Bae et al. [152] developed SLNs loaded with quantum dots PTX and siRNA-bcl2 for the synergistic treatment and in situ lung cancer imaging, paving the way for SLNss to be used as multifunctional and optically traceable nanocarriers for anticancer theragnostics.

## 10. Conclusions

SLNs have been reported to be effective multitasking nanoDDS for cancer treatment since they are demonstrated to be a promising approach to increasing the therapeutic index of the delivered cargo. Currently, multiple SLNs-based formulations are proposed for different cancer treatments with various loading cargos, such as chemotherapeutic drugs and genetic material. Also, modifying the SLNs surface enables effective active targeting against the cells/tissues of interest, increasing the specificity of the treatment and reducing the secondary adverse effects.

Apart from the treatment applications, SLNs also demonstrated great potential as diagnostic tools, openings the possibility of creating multifunctional nanoDDS for theragnostics.

One of the biggest advantages of using SLNs is the possibility of using biocompatible, non-immunogenic, low-cost materials and production methods easily scalable for industrial scale. On the other hand, some limitations related to SLNs, mainly cargo and stability issues, should be overcome. In this sense is expected that the research in the field of SLNs-based nanosystems will keep evolving and that, in the future, more formulations will reach the clinical phases.

## Figures and Tables

**Figure 1 pharmaceutics-15-00831-f001:**
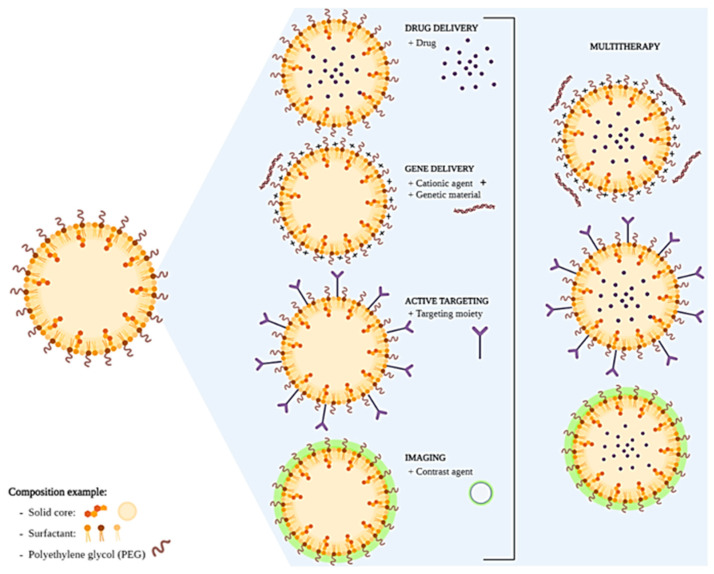
Schematic representation of SLNs development and final applications. Created in BioRender.com, accessed on 24 February 2023.

**Figure 2 pharmaceutics-15-00831-f002:**
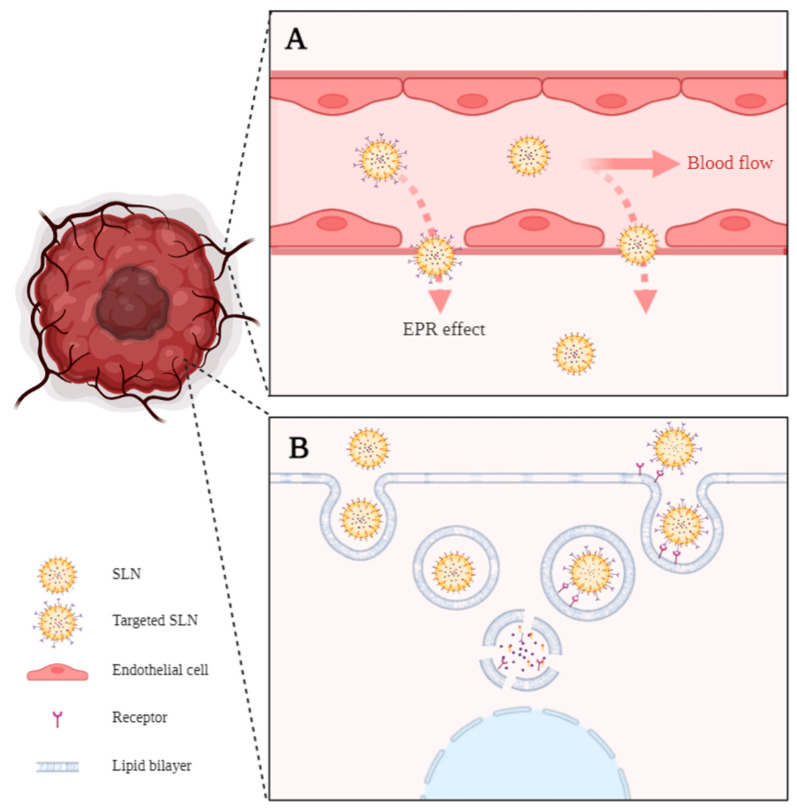
(**A**) Enhanced permeability and retention effect (EPR effect). Passive targeting is explained by the ability of small SLNs to pass through the gaps in the leaky tumor vasculature. (**B**) Passive and active internalization of SLNs by endocytosis. The active internalization implies SLNs functionalized with an antigen and receptor-mediated internalization of the nanoparticle. Created in BioRender.com, accessed on 24 February 2023.

**Figure 3 pharmaceutics-15-00831-f003:**
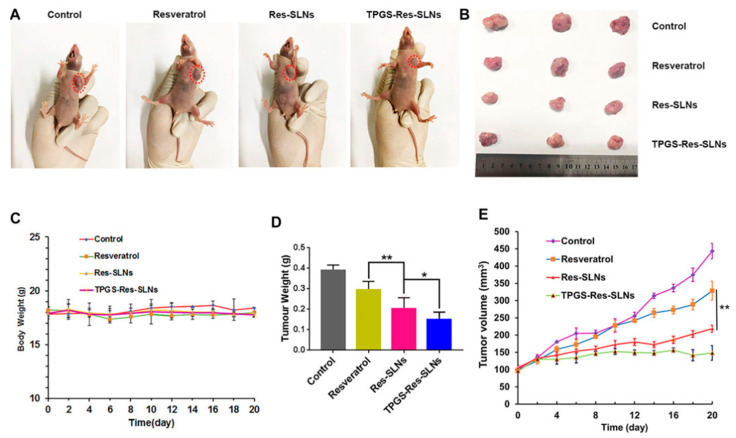
The in vivo efficacies of resveratrol, Res-SLNs, and TPGS-Res-SLNs on mice bearing SKBR3/PR xenografts. (**A**) Representative images of mice on the 16th day in the different treatment groups. (**B**) Digital images of tumors excised from representative mice after the indicated treatments. (**C**) Body weight vs. time curves for mice treated with the indicated formulations. (**D**) Tumor weight of mice in the different treatment groups. (**E**) Tumor volume vs. time curves for mice treated with the indicated formulations. The data represented mean ± SD (*n* = 4). * *p* < 0.05, ** *p* < 0.01. Reprinted with permission of Frontiers Media S.A. under Creative Commons Attribution License (CC BY 4.0) from [71].

**Figure 4 pharmaceutics-15-00831-f004:**
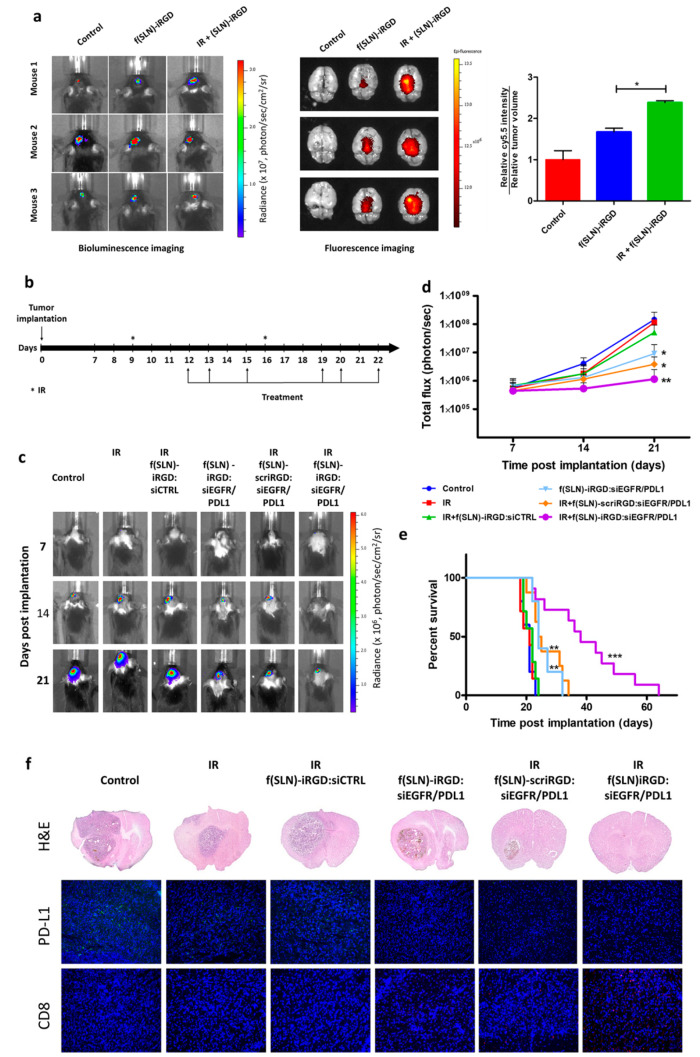
Radiation primes glioblastoma for SLNs targeted delivery. (**a**) Mice bearing GL261-Fluc tumors were irradiated (or not as control) and, three days later, received retro-orbital administration of f(SLNs)-iRGD:Cy5.5 or PBS control. Twenty-four hours post-injection, tumor volume was first evaluated by Fluc imaging (left), and brains were removed and imaged ex vivo for Cy5.5 (middle). The mean fluorescence intensity was calculated and normalized to tumor volume (right; *n* = 3, * *p* < 0.05). (**b**–**f**) Mice bearing GL261-Fluc tumors were irradiated (or not as a control) and retro-orbitally injected with either f(SLNs)-iRGD:siCTRL, f(SLNs)-iRGD:siEGFR/PDL1, or f(SLNs)-scriRGD:siEGFR/PDL1 according to the scheme in (**b**). Tumor growth was monitored weekly by Fluc imaging, and survival was recorded. Images from a representative mouse from each group are shown over time (**c**). Quantification of tumor-associated Fluc radiance intensity with data presented as mean ± SD; * *p* < 0.05 control vs. f(SLNs)-iRGD:siRNA and control vs. IR + f(SLNs)-scriRGD:siRNA; ** *p* < 0.01 control vs. IR + f(SLNs)-iRGD:siRNA by ANOVA (**d**). Kaplan–Meier survival curves are shown (*n* = 5–12); ** *p* < 0.01 control vs. f(SLNs)-iRGD:siRNA; ** *p* < 0.01 control vs. IR + f(SLNs)-scriRGD:siRNA; *** *p* < 0.001 control vs. IR + f(SLNs)-iRGD:siRNA; ** *p* < 0.01 IR + f(SLNs)-scriRGD:siRNA vs. IR + f(SLNs)-iRGD:siRNA; by Mantel–Cox (log-rank) test (**e**). H&E staining and immunohistological analysis using anti-PD-L1 and anti-CD8 antibodies on brain sections of a representative mouse from each group (DAPI, blue; PD-L1, green; and CD8, red) (**f**). Reprinted with permission of American Chemical Society from [145].

**Table 1 pharmaceutics-15-00831-t001:** Commonly used components for SLNs preparation.

Lipids	
**Triglycerides**	Tripalmitin (Dynasan^®^ 116), tristearin (Dynasan^®^ 118), tricaprylate, trimyristin (Dynasan^®^ 114), triolein, trilaurin
**Glycerides**	Glyceril stearate (Precirol^®^ ATO 5), glyceryl palmitostearate, glyceryl dibehenate (Compritol^®^ 888 ATO), behenoyl polyoxyl-8 glycerides (Compritol^®^ HD5 ATO)
**Fatty acids**	Steric acid, palmitic acid, behenic acid, lauric acid, linoleic acid, oleic acid,
**Waxes**	Cetyl palmitate, carnauba wax, beeswax, shellac wax, otoba wax, propolis wax
**Others**	Cholesterol, cocoa butter, hard fat (Gelucire^®^ 43/01, Suppocire^®^ bases, Witepsol^®^ bases), mixture of triceteareth-4 phosphate and ethylene glycol palmitostearate and diethylene glycol palmitostearate (Sedefos^®^ 75), mixture of lauroyl polyoxyl-32 glycerides and PEG 6000 (Gelucire^®^ 59/14), mono and diglycerides and polyoxyl stearate (Gelot^®^ 64)
**Surfactants and Co-surfactants**	
**Phospholipids**	Soy lecithin, egg lectin, phosphatidylcholine
**Polysorbates**	Tween^®^ and Span^®^ derivatives
**Polymers**	Poloxamines, poloxamers, tyloxapol, polyvinyl alcohol (PVA), vitamin E-TPGS, polyoxyethylene-20-cetyl ether, polyoxyethylene glyceryl monostearate, diethylene glycol monoethyl ether, propylene glycol, sodium lauryl sulfate
**Others**	Taurocholic acid sodium salt, taurodeoxycholic acid sodium salt, sodium dodecyl sulfate, cholesteryl oleate, ethanol, butyric acid, polyglyceryl-6 distearate

**Table 2 pharmaceutics-15-00831-t002:** Advantages and disadvantages of the different SLN production techniques.

Technique	Advantages	Disadvantages
**HHPH**	Effective dispersion of particles, reproducible, simple to scale up.	Extremely high energy inputs, high polydispersity
**CHPH**	Suitable for thermo-sensitive drugs	
**Microemulsion**	Low energy inputs, flexibility of interphase, simple to scale-up	Low lipid content, exposure to high temperatures
**Double microemulsion**	Higher particle size compared to microemulsion	
**HSH**	No use of organic solvents, no use of high amounts of surfactants, low production cost	High polydispersity, poor encapsulation efficiency
**SEE**	Suitable for thermo-sensitive drugs, small particle diameter, simple to scale-up	Toxicity due to organic solvents, low lipid content, possible aggregation
**SED**	Pharmaceutically accepted organic solvent, low polydispersity, small particle diameter, simple to scale up	Low lipid solvent, possible organic solvent toxicity
**SI**	Pharmaceutically accepted organic solvent, simple to scale up, highly efficient, and versatile	Difficult to remove solvent, low lipid content
**Coacervation**	Suitable for lipophilic drugs, no organic solvent, monodispersity, simple to scale-up	Not suitable for pH-sensitive drugs
**MAMT**	Controlled microwave heating, low energy inputs	Problems to scale-up

Abbreviations: Hot High-Pressure Homogenization (HHPH), Cold High-Pressure Homogenization (CHPH), High Shear Homogenization (HSH), Solvent Emulsification-Evaporation (SEE), Solvent Emulsification-Diffusion (SED), Solvent Injection (SI), Microwave-Assisted Microemulsion Technique (MAMT).

## Data Availability

Not applicable.

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
