# Peer review of "Solid Lipid Nanoparticles: Multitasking Nano-Carriers for Cancer Treatment"

_pharmaceutics, 2023, doi:10.3390/pharmaceutics15030831_

Round 1
Reviewer 1 Report
It was a manuscript about the application of SLN nanocarrier for cancer therapy. here are some comments on this study that should be considered before publication:
1- Why do you use “Multitask” in the title of the manuscript?
2- Please improve the quality of figure 1.
3- There are grammatical mistakes in the text that should be corrected.
4- Section 3.3 needs to be rewritten.
5- The title of heading 4 is not compatible with its text, please check and correct it.
6- Please use the same format for all the tables.
7- In tables 4 and 5, please mention the results of studies instead of phase.
8- The caption of table 4 is not available.
9- Please rewrite the conclusion section.
10- Please add a new heading and mention the limitations and future perspectives related to the application of SLN.
Author Response
The authors acknowledge all the suggestions made for the improvement of the overall quality of the paper. All the question was answered point-by-point. Please see the attached file.

Reviewer 2 Report
The authors have provided a comprehensive overview of the potential benefits of using solid lipid nanoparticles (SLN) in cancer treatment, highlighting their ability to deliver different types of therapeutic cargo and act on different mechanisms of cancer cells, as well as their potential for personalized medicine. However, it is also important to note that while promising, the development and application of SLN in cancer treatment are still in their early stages, and more research is needed to fully realize its potential. Additionally, while SLN has the potential to reduce the side effects of conventional cancer treatments, it is important to thoroughly evaluate and monitor the safety and efficacy of SLN in clinical trials.
Some comments to the authors:
1. In the introduction, there is no mention of the problems or problems that still need to be solved when using solid lipid nanoparticles (SLN) to treat cancer.
2. This introduction is missing a clear definition or explanation of what SLN is and the context for why it is being explored for drug and gene delivery as well as diagnostic tools.
3. The advantages of SNL synthesis can vary depending on the type of drug, desired particle size, and other factors. Would it be possible to highlight the importance of choosing the right production method based on these factors and the desired application.
4. Despite the advantages, solid lipid nanoparticles (SLNs) also have certain limitations and criticisms, such as limited stability. High manufacturing costs; limited drug loading capacity; Limitations on administration routes, and toxicity issues, SLNs are composed of lipids; please include such limitations.
5. There are various limitations to It is worth noting that these limitations are actively being researched, and new strategies are being developed to overcome them, but at present, these remain challenges for widespread adoption of SLN technology in cancer treatment.
The conclusion in the text does not mention the potential limitations or challenges of using SLN for cancer treatment, such as potential toxicity, difficulties in controlling the release of the payload, or issues with scalability and manufacturing. Additionally, the conclusion does not mention the need for further research and testing to determine the safety and efficacy of SLN in different cancer types and treatment regimens.
Author Response

(The authors gave the same response as above.)

Reviewer 3 Report
This manuscript talks about, various aspects related with SLN, including composition, production methods and administration routes, as well as to show the most recent studies about use of SLN for cancer treatment.
The comments and observation to improve the review are as follows:
1. Table 1: Update the list of components with novel materials (Brand as well as common name) introduce in last two decade, may classify as derivative if necessary. (Few listed in Table 3)
2. Figure 1: What does mean by selection of lipid and selection of surfactant? There is just list of ingredients that are summarized in Table 1. Information is duplicating. Please include factors used behind selection of lipids and surfactants in figure.
3. Line 78: SLN can be modulated, modulated? Formulated or prepared is more appropriate word. Please check for all such word if any in manuscript.
4. Line 93: Up to know, different methods?... Rewrite such sentences for proper meaning.
5. Add process flow diagrams in form of image for 2.1,2.2 and so on processes.
6. Table 2, include examples, case study and cite each one separately. Update table as it very basic information. Name few equipment work on these techniques (Lab and Larges scale)
7. Suggestion: Modify the heading: 3. Solid Lipid Nanoparticles Applications and Administration Routes to simply SLNs in cancer treatment (Different type of cancer and how SLN improve therapy), give general outline.
8. Is heading 4 and 5 are ok. Please review the sequence of headings.
9. 7. SLN for gene delivery or gene therapy?
10. Present conclusion is like summary work. Please make in one small paragraph.
11. What should be included?
a. Challenges for scale up
b. Characterization and optimization of SLNs
c. Marketed formulations and patents
d. Clinical trails studies, completed and under progress
e. Add more citation of 2022, at present only 2 are there. Current literature should be included.
12. What should be omitted?
a. Cut short introduction, and focus on recent development
Author Response

(The authors gave the same response as above.)

Round 2
Reviewer 1 Report
-
Reviewer 2 Report
No further comments to authors
Reviewer 3 Report
I am satisfied with the revision. Thank you.